# Diet and Depression during Peri- and Post-Menopause: A Scoping Review Protocol

**DOI:** 10.3390/mps6050091

**Published:** 2023-10-02

**Authors:** Alexandra M. Bodnaruc, Miryam Duquet, Denis Prud’homme, Isabelle Giroux

**Affiliations:** 1School of Nutrition Sciences, Faculty of Health Sciences, University of Ottawa, Ottawa, ON K1N 6N5, Canada; abodn049@uottawa.ca (A.M.B.); mduqu016@uottawa.ca (M.D.); 2Université de Moncton, Moncton, NB E1A 3E9, Canada; denis.prudhomme@umoncton.ca

**Keywords:** depression, depressive disorders, dietary patterns, food groups, foods, nutrients, peri-menopause, post-menopause, women, scoping review

## Abstract

The aim of the proposed scoping review is to describe and summarize studies assessing the associations between diet-related variables and depression in peri- and post-menopausal women. Studies examining the associations between diet-related variables and mental health indicators in women undergoing menopausal transition or in the post-menopausal period will be systematically retrieved via Medline, EMBASE, PsycINFO, Web of Science, and Scopus databases. All articles identified through the database searches will be imported into Covidence. Following the removal of duplicates, two authors will independently perform title and abstract screening, as well as full-text assessment against eligibility criteria. Data will be extracted using tables developed for observational and experimental studies. The methodological quality of randomized trials, cohort and cross-sectional studies, and case–control studies, will be assessed using the Cochrane risk-of-bias (RoB-2) tool, the NHLBI Quality Assessment Tool for Observational Cohort and Cross-Sectional Studies, and the NHLBI Quality Assessment Tool for Case–Control studies, respectively. Data extraction tables will be used to produce two tables summarizing the main characteristics and findings of the studies included in the review. In the proposed review, we will systematically identify and summarize the currently available evidence on the association between diet-related variables and depression in peri- and post-menopausal women. To our knowledge, this is the first review focusing on this subgroup of the population. Protocol registration: osf.io/b89r6.

## 1. Introduction

Depression is a highly distressing and often chronic neuropsychiatric disorder characterized by persistent low mood and/or anhedonia, along with alterations in emotion regulation (e.g., excessive feelings of guilt, unworthiness), cognitive abilities (e.g., decreased concentration, memory), and physiological functions (e.g., dysregulated sleeping patterns and appetite, hypo- or hyper-locomotion) [1]. According to the Global Burden of Diseases, Injuries, and Risk Factors Study 2019, depression is the most widespread mental health disorder, affecting over 250 million individuals of all ages and placing second among the leading causes of disability worldwide [2]. Individuals with depression have a higher risk of several psychiatric [3,4,5] and cardiometabolic [6,7,8] disorders and experience significant impairments in their ability to perform activities of daily living, as well as to fulfill their social, familial, and professional functions [9,10,11,12], all of which contribute to reducing their quality of life [13] and have critical societal repercussions [14,15]. 

Depression is known to disproportionately affect women, with studies conducted across a wide range of nations and cultures reporting a twofold higher prevalence in women than men [16]. Sex-related differences have also been described regarding symptom prevalence and severity, treatment responsiveness, and associated psychiatric comorbidities [17,18,19,20,21,22]. Comparisons between women and men with major depressive disorder and/or persistent depressive disorder have consistently revealed a higher prevalence of somatization, sleep disturbances, appetite dysregulation, and weight gain in women [17,18,19,20]. There have also been reports of greater symptom severity [17,20] and higher prevalence of atypical depression features [17,18,19,20], generally associated with treatment resistance and chronicity [21,22], in women as compared to men. 

Biological factors contributing to females’ higher vulnerability to depression mainly arise from the effects of chromosomal and gonadal sexual differentiation, including the resulting variations in lifelong sex hormone exposure [23,24]. In addition to their reproductive functions, studies have reported that estrogens could modulate stress responses and may be involved in mood regulation by binding to β-estrogenic receptors expressed in brain regions such as the hypothalamus, the hippocampus, and the amygdala, thereby interacting with serotoninergic, noradrenergic, and dopaminergic neurotransmitter systems [25,26]. As is consistent with estrogen’s neuroendocrine functions, sex-related differences in the prevalence of depression become apparent starting at puberty [16]. In addition, in women, the prevalence and the severity of depressive symptoms have been shown to rise in parallel with reproductive stages characterized by decreased or highly variable estrogen levels, such as the late luteal phase of the menstrual cycle, the post-partum period, as well as peri- and post-menopause [27]. 

The menopause transition is a time when women experience a range of endocrine changes that ultimately lead to the end of their reproductive capacities. Women undergoing the menopausal transition are two to five times [28,29,30,31,32,33,34,35] more likely to develop depression or to experience depressive symptoms than pre-menopausal women, and present with a distinct symptomatologic signature [36,37]. In addition to changes in sex hormone levels, such as estrogen, progesterone, and testosterone, the menopause transition has been associated with higher pro-inflammatory and oxidative biomarkers [38,39,40], which can, among others, increase the risk of depression [41,42,43,44,45,46,47]. In contrast with premenopausal women, persistent depressive mood is less frequently experienced by women undergoing menopausal transition, who rather display rapid mood changes along with higher levels of irritability, paranoia, and fatigue [36,37]. Further complicating this picture, in peri- and post-menopausal women, depression is often exacerbated by vasomotor symptoms [35,48,49], and women have shown an overall poor response to common antidepressants when compared with pre-menopausal women [36,50,51,52,53]. 

In recent years, in an effort to improve prevention and to complement currently available treatments, an astounding body of knowledge has been generated on the relationship between diet and depression. The overall diet, as well as individual components of the diet, interact with physiological systems displaying functional alterations that are suggested to contribute to the pathogenesis of depression [54], including monoamine metabolism [55,56], neurotrophic factor synthesis [57,58], the hypothalamic–pituitary–adrenal axis [59,60], oxidative stress [44,45,46,47], and inflammation [41,42,43]. While numerous systematic reviews have addressed the relationship between diet and depression in the general population [61,62,63,64,65,66,67,68,69,70,71,72], less attention has been dedicated to female populations, who are known to be at a higher risk of depression. In particular, there is, to our knowledge, no systematic synthesis of this relationship among peri- or post-menopausal women. To address this gap in the literature, we will conduct a scoping review with the aim of describing and summarizing studies assessing the associations between diet-related variables and depression in peri- and post-menopausal women.

## 2. Methods

This protocol has been registered within the Open Science Framework platform (osf.io/b89r6). A preliminary search of PROSPERO, Cochrane Database of Systematic Reviews, and the Joanna Briggs Institute Database of Systematic Reviews and Implementation Reports was conducted, and no existing or ongoing reviews with similar aims were identified. This protocol is reported following the Preferred Reporting Items for Systematic Reviews and Meta-Analyses Protocols (PRISMA-P) statement [73] (Appendix A), and was informed by Arksey and O’Maley’s [74] framework for conducting scoping reviews as well as Peters et al.’s [75] guidelines for scoping reviews. 

### 2.1. Step 1: Identifying the Search Questions

This scoping review aims to answer the following question: “To date, what is the available evidence on diet-related variables and depression in peri- and post-menopausal women?”. The sub-questions are: What are the characteristics of available evidence on diet-related variables and depression in peri- and post-menopausal women?What are the main findings of available evidence on diet-related variables and depression in peri- and post-menopausal women?What are the main research gaps on the topic of diet-related variables and depression in peri- and post-menopausal women?

### 2.2. Step 2: Identifying Relevant Studies

Studies examining the associations between diet-related variables and depression in peri- and post-menopausal women will be systematically retrieved via Medline, EMBASE, PsycINFO, Web of Science, and Scopus. The search strategy was developed with the guidance of an experience professional health science information specialist. To ensure a comprehensive search, the search terms included both database-specific subject headings and keywords. The search will be conducted from the inception of each database to the date of the last search. Languages will be restricted to English and French. The search strategies for Medline as well as EMBASE, PsycINFO, Web of Science, and Scopus are presented in Table 1 and Appendix A, respectively. Backward citation tracking of all included articles will be carried out to identify any other pertinent articles.

### 2.3. Step 3: Study Selection

To be eligible for inclusion in this review, studies will be required to meet the criteria described below regarding the types of participants, exposures, interventions, comparators, outcomes, and study designs.

#### 2.3.1. Type of Participants

The scoping review will focus on studies including (1) healthy peri- and post-menopausal women or (2) peri- and post-menopausal women with diagnoses of primary major or persistent depressive disorder prior to enrollment in the study. No restrictions will be applied as to participants’ age or race. Only studies focusing exclusively on women with chronic health conditions other than those mentioned above, women on sex hormone replacement therapy, and women who underwent hysterectomies will be excluded. 

#### 2.3.2. Type of Exposures and Interventions

The exposures and interventions of the included studies may include a wide array of diet-related variables. Diet-related variables will only be excluded if they are considered unusual, linked to an underlying health condition or surgical procedure, or are unlikely to be found in foods in their unaltered forms. As such, while studies using macronutrients, vitamin, mineral, and phytonutrient (e.g., phenolic compounds, non-digestible carbohydrates) supplements will be considered eligible, those involving herbal supplements (e.g., Ginkgo biloba, ginseng, St. John’s Wort) or any pharmaceutical agents (e.g., Liraglutide, Naltrexone-bupropion, Orlistat, etc.) aimed at modifying eating behaviors, food intake, and/or nutrient metabolism will be excluded. No restrictions will be applied as to dietary intake assessment methods in observational studies, nor intervention duration in experimental studies. 

#### 2.3.3. Type of Comparators

Experimental studies will be considered eligible if the dietary intervention of interest is compared to (i) a placebo, (ii) another dietary intervention, or (iii) no intervention.

#### 2.3.4. Type of Outcomes

Outcomes will be limited to unipolar major and persistent depressive disorder with or without current treatment, as well as to depressive symptoms. No restrictions will be applied as to the methods and tools used to assess depression and depressive symptoms. Studies focusing exclusively on depressive symptoms as part of the symptomatology of another physical (e.g., hypothyroidy, anemia, cardiometabolic disorders, etc.) or mental health disorder (e.g., schizophrenia, eating disorders, personality disorders, declined cognitive functions, etc.) will be excluded.

#### 2.3.5. Type of Study Designs

Primary experimental (i.e., randomized controlled parallel and crossover trials with individual and cluster randomization) and observational (i.e., cohort, case–control, and cross-sectional studies) studies will be considered eligible. Preclinical trials, case studies, and case series will be excluded.

#### 2.3.6. Selection of Studies

All records identified through the database search will be imported into Covidence and duplicates will be removed. All remaining records will be screened against the title and abstract independently by two authors (A.M.B. and M.D.). At this stage, articles will only be excluded if it can be clearly determined by the title or abstract that they did not meet the inclusion criteria. Every discrepancy in eligibility will be discussed between the two assessors and articles for which consensus cannot be reached will be moved to the full-text screening stage on the premise that the articles’ titles and abstracts may contain insufficient information to determine their eligibility. Articles deemed eligible based on the title and abstract will undergo full-text reviewing by the same two assessors, with discrepancies to be resolved through discussion and consensus.

### 2.4. Step 4: Charting the Data

One author will perform the data extraction from papers of eligible studies. Two data extraction tables, one for observational studies and one for experimental studies, were built and will be piloted with a small number of articles. Adjustments will be made prior to and during the data extraction process, as necessary. The preliminary data extraction fields are as follows: (i) authors, (ii) publication year, (iii) protocol registration number (where applicable), (vi) protocol publication reference (where applicable), (v) study location, (vi) study design, (vii) study duration (where applicable), (viii) participant recruitment type, (ix) type of randomization (where applicable), (x) number of participants, (xi) age, (xii) menopause stage(s), (xiii) dietary variable name, (xiv) tools used to assess dietary variable, (xv) variable type, (xvi) outcome name, (xvii) tools used to assess depression or depressive symptoms, (xviii) quantitative results with description of statistical analysis type, and (xix) statistical analysis adjustments (where applicable).

### 2.5. Step 5: Collating, Summarizing, and Reporting Results

Data extraction tables will be used to produce two tables summarizing the main characteristics and findings of the studies included in the review. In the results section, a descriptive summary of the included studies’ general characteristics will be provided and divided into four sections: types of studies, types of participants, types of exposures and interventions, and types of outcomes. The main findings of the included studies will be summarized according to the type of nutritional exposure/intervention. Research gaps will be addressed in the discussion section of the manuscript. 

### 2.6. Step 6: Methodological Quality Appraisal

Risk-of-bias assessment is not a mandatory step in scoping reviews as they are typically viewed as a starting point for more subject-targeted systematic reviews, which require methodological quality assessment [76,77,78]. Scoping reviews can also sometimes include large numbers of studies, making quality assessment an arduous step [76]. Despite this, quality assessment can strengthen the conclusions of a scoping review and might be particularly pertinent when its results do not support the need for conducting systematic reviews on the topic [76]. For these reasons, quality assessment of eligible studies will be undertaken in the proposed scoping review. Quality assessment will be performed by AB.

The risk of bias in randomized trials, cohort and cross-sectional studies, and case–control studies will be assessed using the revised Cochrane risk-of-bias (RoB-2) tool [78]; the National Heart, Lung, and Blood Institute (NHLBI) Quality Assessment Tool for Observational Cohort and Cross-Sectional Studies [79]; and the NHLBI Quality Assessment Tool for Case–Control studies [80], respectively.

The Cochrane RoB-2 tool assesses five bias domains known to affect the results of randomized and quasi-randomized trials, namely, (i) bias arising from the randomization process, (ii) bias due to deviations from intended interventions, (iii) bias due to missing outcome data, (iv) bias in the measurement of the outcomes, and (v) bias in the selection of the reported results. Each of these domains contains guiding questions, which can be answered by “yes”, “probably yes”, “no”, “probably no”, or “no information”. Based on assessors’ answers to the guiding questions, an overall judgement of either “low risk of bias”, “some concerns”, or “high risk of bias” will be reached for each domain. Using the judgements reached for each domain, the study itself will be rated as:(i)Being at low risk of bias when all domains are rated as such;(ii)Raising some concerns when at least one domain is rated as such, but no domain is rated as being at high risk of bias;(iii)Being at high risk of bias when at least one domain is rated as such, or when multiple domains are rated as raising some concerns.

The NHLBI Quality Assessment Tool for Observational Cohort and Cross-Sectional Studies and the NHLBI Quality Assessment Tool for Case–Control Studies consist of 14 and 12 items, respectively, assessing common sources of bias in observational studies, namely, (i) bias from participants’ recruitment or selection methods and sample size; (ii) bias in the measurement of the exposures; (iii) bias in the measurement of the outcomes; (iv) bias due to the handling of potential confounders; and (v) bias in the selection of the reported results. Each item included in the NHLBI Quality Assessment Tools can be answered by “yes”, “no”, or “no information”. Cohort and cross-sectional studies will be considered as being at a low risk of bias when the answer to ≥13 items is yes; at a moderate risk of bias when the answer to 10, 11, or 12 items is yes; and at a high risk of bias when the answer to <10 items is yes. Case–control studies will be considered to be at a low risk of bias when the answer to ≥11 items is yes; at a moderate risk of bias when the answer to 8, 9, or 10 items is yes; and at a high risk of bias when the answer to <8 items is yes.

## 3. Discussion

In the proposed review, we will systematically identify and summarize the currently available evidence on the association between diet-related variables and depression in peri- and post-menopausal women. To our knowledge, this is the first review focusing on this subgroup of the population. Potential limitations of the proposed methods include a higher risk of bias due to publication language restriction, the fact that we will not seek out unpublished reports, and the absence of duplicate data extraction and risk-of-bias assessment. 

Upon completion, this scoping review will be the most comprehensive and up-to-date compilation of existing evidence concerning the relationship between diet-related variables and depression in peri- and post-menopausal women. It will serve as a pivotal point of reference, directing future research and offering a valuable resource for health professionals in the fields of nutrition, psychiatry, and women’s health. 

## Figures and Tables

**Table 1 mps-06-00091-t001:** Medline (through Ovid) search strategy for identifying studies on diet-related variables and mental health during the menopause transition and post-menopause.

	Search Terms
Theme #1 Diet	exp diet/ exp nutritive value/ exp hunger/ food/ or dairy products/ or dietary carbohydrates/ or dietary fats/ or dietary fiber/ or dietary proteins/ or dietary supplements/ or eggs/ or fast foods/ or fruit/ or meals/ or meat/ or micronutrients/ or nuts/ or seeds/ or vegetables/diet*.mp. nutritional or physiological phenomena/ eating/ feeding behavior/ appetite regulation/ diet*.mp nutriti*.mp food.mp eat*.mp energy intake.mp (macronutrient* or micronutrient* or nutrient*).mp 1 OR 2 OR 3 OR 4 OR 5 OR 6 OR 7 OR 8 OR 9 OR 10 OR 11 OR 12 OR 13 OR 14
Theme #2 Depression	16. exp depressive disorder/ 17. depression/ 18. mood disorders/ 19. (depression* or (depressive adj3 (condition* or disorder* or symptom*))).mp 20. 16 OR 17 OR 18 OR 19
Theme #3 Menopause	21. exp menopause/ 22. menopaus*.ti,ab,kw. 23. premenopaus*.ti,ab,kw. 24. perimenopaus*.ti,ab,kw. 25. postmenopaus*.ti,ab,kw. 26. 21 OR 22 OR 23 OR 24 OR 25
Combining search terms	27. 15 AND 20 AND 26
Language	28. limit 27 to (english or french)
Other limits	29. limit 28 to humans
	30. limit 29 to females

## Data Availability

Not applicable.

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
