# Peer review of "Diet and Depression during Peri- and Post-Menopause: A Scoping Review Protocol"

_mps, 2023, doi:10.3390/mps6050091_

Round 1

Reviewer 1 Report

The manuscript indicates the gap in the literature about a scoping review that summarizes studies assessing the associations between diet-related variables and mental health indicators in peri- and post-menopausal women. In addition, it describes the methodology that the authors will carry out to do this scoping review.

The topic is interesting and the methodology is clear. However, a few minor issues should be taken into account:
- First, although the title and the search strategy talk about "mental health",  the introduction and discussion sections only consider "depression". Therefore, other disorders included in the selection criteria should be justified and discussed (e.g. anxiety symptoms, psychological stress, negative body image, and body dysmorphic disorder).

- Second, in the Methods section, the  first sub-question is not clear. I mean, what do you want to say with "the characteristics of available evidence"?

- Third, why does your selection include only articles published from 1993? I consider that a review should be carried out without a time limit.

Author Response

Reviewer #1

The manuscript indicates the gap in the literature about a scoping review that summarizes studies assessing the associations between diet-related variables and mental health indicators in periand post-menopausal women. In addition, it describes the methodology that the authors will carry out to do this scoping review. The topic is interesting and the methodology is clear. However, a few minor issues should be taken into account:

- First, although the title and the search strategy talk about "mental health", the introduction and discussion sections only consider "depression". Therefore, other disorders included in the selection criteria should be justified and discussed (e.g. anxiety symptoms, psychological stress, negative body image, and body dysmorphic disorder).

Based on feedback from both reviewers of this protocol paper, we have decided to limit the review to depression only. Changes have been made at lines 164-182 and throughout the revised manuscript.

- Second, in the Methods section, the first sub-question is not clear. I mean, what do you want to say with "the characteristics of available evidence"?

We refer to the characteristics of studies, such as study design, year of publication, country, etc. We have added a clarification at line 108 of the revised manuscript.

- Third, why does your selection include only articles published from 1993? I consider that a review should be carried out without a time limit.

We have changed this criterion and will search articles published from the inception of the database to the date of the last search. Please see changes at lines 123-124 of the revised manuscript.

Reviewer 2 Report

Dear authors,

This study examined to diet and mental health during peri- and post-menopause: a scoping review protocol. Although this is the first review focusing on this population subgroup (peri- and post-menopausal women), it has major concerns in this study.

Moreover, this manuscript is clumsy, and the result table and interpretation of the results are beginner level. And, this study has serious flaws in terms of rationale, methods, understandings discussed (no Discussion section) in the manuscript.

Major concerns

(1)   Why do not use Pubmed and Scopus database in this study? This makes serious bias of the results.

(2)   You have to describe the Results and Conclusion section. In this study, there has no Results and Conclusion section.

(3)   Moreover, based on the PRISMA-P methods, you have to show the results of Meta-Analyses and Charting.

Minor concerns

(1) You have to add some paragraphs in Introduction section about backgrounds of diet variables, mental health except depression factor (it was already described), and detailed characteristics of peri- and post-menopause population.

(2) (Line 101-102) You should revise it.

(Before) Arksey & O'Maley's (2005)[93] framework for conducting scoping reviews as well as Peters et al.'s (2015) [94] guidelines for scoping reviews.

(After) Arksey & O'Maley's [93] framework for conducting scoping reviews as well as Peters et al.'s [94] guidelines for scoping reviews.

- Please add limitations of this study in Limitation section, and add application in diet and mental health field from this study.

- I recommend that this manuscript should be edited by an English professional editor for more readable. There are several typo and grammatical errors.

Author Response

Reviewer #2

Dear authors,

This study examined to diet and mental health during peri- and post-menopause: a scoping review protocol. Although this is the first review focusing on this population subgroup (peri- and postmenopausal women), it has major concerns in this study. Moreover, this manuscript is clumsy, and the result table and interpretation of the results are beginner level. And, this study has serious flaws in terms of rationale, methods, understandings discussed (no Discussion section) in the manuscript.

Major concerns

(1) Why do not use Pubmed and Scopus database in this study? This makes serious bias of the results.

As suggested, we will add Scopus to our search strategy. Please see changes at line 120 of the revised manuscript. PubMed and Medline are very similar and are rarely both searched in systematic and scoping reviews. We are searching Medline, which includes over 95% of PubMed articles.

(2) You have to describe the Results and Conclusion section. In this study, there has no Results and Conclusion section.

This a scoping review protocol, therefore there is no results and conclusion section required. The results of this scoping review protocol will be published in a separate manuscript.

(3) Moreover, based on the PRISMA-P methods, you have to show the results of Meta-Analyses and Charting.

The PRISMA-P checklist is developed for systematic reviews, while our review is a scoping review. While we used PRISMA-P as a guide for reporting this protocol, not all PRISMA-P items apply to scoping reviews. The results of scoping review are generally presented narratively, without meta-analyses. Details on data charting are provided at lines 202-213.

Minor concerns

(1) You have to add some paragraphs in Introduction section about backgrounds of diet variables, mental health except depression factor (it was already described), and detailed characteristics of peri- and post-menopause population.

Based on feedback from both reviewers of this paper, we have decided to limit the review to depression only. Changes have been made at lines 164-182 and throughout the revised manuscript.

(2) (Line 101-102) You should revise it. (Before) Arksey & O'Maley's (2005)[93] framework for conducting scoping reviews as well as Peters et al.'s (2015) [94] guidelines for scoping reviews. (After) Arksey & O'Maley's [93] framework for conducting scoping reviews as well as Peters et al.'s [94] guidelines for scoping reviews.

The revision has been made as suggested at lines 101-102 of the revised manuscript.

(3) Please add limitations of this study in limitation section, and add application in diet and mental health field from this study.

Additions have been made at lines 270-285 of the revised manuscript

Round 2

Reviewer 2 Report

The authors have extensively improved the quality of the manuscript by addressing the reviewers' comments and it can now be accepted for publication.

The authors have extensively improved the quality of the manuscript by addressing the reviewers' comments and it can now be accepted for publication.

Author Response

We would like to thank this reviewer for his feedback that has enhanced the quality of our manuscript. 

Kind regards, 

Alexandra Bodnaruc